# Analysis of the RNA Editing Sites and Orthologous Gene Function of Transcriptome and Chloroplast Genomes in the Evolution of Five *Deutzia* Species

**DOI:** 10.3390/ijms241612954

**Published:** 2023-08-19

**Authors:** Hongyu Cai, Yachao Ren, Juan Du, Lingyun Liu, Lianxiang Long, Minsheng Yang

**Affiliations:** 1Forestry College, Hebei Agricultural University, Baoding 071000, China; 2Hebei Key Laboratory for Tree Genetic Resources and Forest Protection, Baoding 071055, China; 3Shijiazhuang Botanical Garden, Shijiazhuang 050299, China

**Keywords:** chloroplast genome, *Deutzia* species, gene function, orthologous genes, phylogenetic analysis, RNA editing

## Abstract

In this study, the chloroplast genomes and transcriptomes of five *Deutzia* genus species were sequenced, characterized, combined, and analyzed. A phylogenetic tree was constructed, including 32 other chloroplast genome sequences of Hydrangeoideae species. The results showed that the five *Deutzia* chloroplast genomes were typical circular genomes 156,860–157,025 bp in length, with 37.58–37.6% GC content. Repeat analysis showed that the *Deutzia* species had 41–45 scattered repeats and 199–201 simple sequence repeats. Comparative genomic and pi analyses indicated that the genomes are conservative and that the gene structures are stable. According to the phylogenetic tree, *Deutzia* species appear to be closely related to *Kirengeshoma palmata* and *Philadelphus*. By combining chloroplast genomic and transcriptomic analyses, 29–31 RNA editing events and 163–194 orthologous genes were identified. The *ndh*, *rpo*, *rps*, and *atp* genes had the most editing sites, and all RNA editing events were of the C-to-U type. Most of the orthologous genes were annotated to the chloroplast, mitochondria, and nucleus, with functions including energy production and conversion, translation, and protein transport. Genes related to the biosynthesis of monoterpenoids and flavonoids were also identified from the transcriptome of *Deutzia* spp. Our results will contribute to further studies of the genomic information and potential uses of the *Deutzia* spp.

## 1. Introduction

*Deutzia* (family Saxifragaceae, subfam. Hydrangeoideae) is a genus of deciduous shrubs comprising approximately 60 species worldwide. China is the main distribution area of the genus *Deutzia*, with 53 species (two of which were introduced or naturalized), one subspecies, and 19 varieties in China, accounting for approximately 80% of the whole genus; *Deutzia* spp. are found in all provinces of China [1,2,3]. In northern China, *Deutzia* spp. are ornamental plants that appear in early summer; these are also valuable supplementary nectar source plants in late spring and early summer [2,3]. *Deutzia* roots, leaves, and fruits can be used medicinally due to their antipyretic, diuretic, insecticidal, and bone-setting effects [4]. The medicinal components of *Deutzia* spp. are mainly iridoids and flavonoids contained in the leaves. Iridoids are widely found in traditional Chinese medicine, including Gentianaceae, Rubiaceae, Labiatae, and Scrophulariaceae. Iridoids have a variety of physiological activities, such as gallbladder and stomach protection, as well as hypoglycemic, antibacterial, and anti-inflammatory properties [5]. Because *Deutzia* spp. may have potential as a pharmaceutical raw material, its medicinal value is worthy of further exploration. Therefore, a better understanding of the development, evolution, and internal molecular mechanisms of *Deutzia* spp. is necessary.

According to morphological classification, the genus *Deutzia* is divided into three sections, *Deutzia* (four subsections), *Mesodeutzia* (three series), and *Neodeutzia* [2,6]. In recent years, molecular biology research on *Deutzia* species has revealed several important findings. The results of SRAP molecular marker studies conducted by J. Lu et al. on the genetic diversity of *D. baroniana* and *D. glabrata* showed that both species have high levels of genetic diversity [7,8]. The molecular phylogeny of *Deutzia* has also been explored, with a focus on ITS, 26S and *matK*, *rbcL*, and *trnL-F* genes [9]. However, studies about *Deutzia* are still in the initial stages. Classification of the *Deutzia* genus has been based mainly on morphology [2,6], especially the relationship between the main lineages and lower groups. Therefore, more in-depth research into the molecular biology and gene construct of the *Deutzia* genus is warranted.

The chloroplast is an important semi-autonomous genetic organelle. Photosynthesis occurs within chloroplasts, which are the energy sources that play a key role in plant evolution. Future studies and exploitation of chloroplast function depend on a comprehensive understanding of the chloroplast genome and its role in evolution. The coding genes in plant plastids originate from cyanobacteria and are transferred from the endosymbiont genome to the ‘host’ nucleus [10,11,12]. Previous studies determined that although the entire structure and genetic composition of plant chloroplast genomes are relatively conserved, many mutations still exist, and some genes in the chloroplast genome are lost or transferred during the evolutionary process. For example, some angiosperms have lost genes, such as *ycfl*, *rpl23*, *infA*, and *rpsl6* [13], while others have dropped *ycfl*, *rpl23*, *infA*, and *rpsl6*. Moreover, all of the *ndh* gene family was lost during evolution in *Pinus* spp. [14]. Some parasitic plants have lost most of their photosynthesis-related genes due to the degradation of their photosynthetic function [15]. Thus, plant chloroplast genomes appear to be influenced by environmental selection pressures in the process of evolution, which in turn lead to differences in the evolutionary direction. The gene transfer event between chloroplast genes and nuclear genomes is an important process in evolution, and this kind of transfer has been ongoing. For example, 18 and 33 kb chloroplast DNA were found inserted in the nuclear genomes of tobacco and rice, respectively [16,17]. More than 11 Kb DNA insertions were also detected in *Arabidopsis* [18]. Huang et al. used a neomycin phosphotransferase to demonstrate that a gene was transferred from the chloroplast to the nucleus [19]. However, the mechanism by which plastid DNA sequences are transferred has yet to be clarified, and little is known about how the nuclear genome is integrated.

RNA editing is a functionally significant post-transcriptional regulatory process [20] that manifests mainly as nucleotide insertions/deletions or substitutions [21]. It is essential for chloroplast development in advanced plants. Many RNA editing positions have been successively identified in plant chloroplast genomes. For example, maize possesses 26 RNA editing site positions [22], tobacco possesses 31 [23], and *Hevea brasiliensis* has 51 [24]. Most types of RNA editing do not change the nucleotide base-pairing properties except in A-to-I and C-to-U editing. In plants, C-to-U editing occurs more frequently in chloroplast and mitochondrial genes [22,23,25,26,27]. The expression level of a gene is correlated with C-to-U editing levels. Cytosines with high editing levels are more conserved at the DNA level [28,29]. Correlation factors of C-to-U conversion, such as MORFs and PPR proteins, are also involved in tea and rice albinism [30,31,32]. Other types include U-to-C substitutions, which are much less frequent [33]. The C-to-U editing group is formed by natural selection pressure, and most non-synonymous C-to-U editing is adaptive. The RNA editing machinery can be selected and actively maintained, which may have far-reaching impacts on modified RNAs [29].

To further study the evolution of *Deutzia* spp. and the utilization of *Deutzia* plant resources, we collected the plant material from five common wild-type *Deutzia* species in Hebei Province, including *Deutzia scabra*, *Deutzia grandiflora*, *Deutzia parviflora*, *Deutzia hamata*, and *Deutzia hypoglauca*. In this study, the chloroplast genomes and transcriptomes of *D. scabra*, *D. grandiflora*, *D. parviflora*, *D. hamata*, and *D. hypoglauca* were sequenced and characterized and then compared with the other 32 Hydrangeoideae species. The transcriptomes were combined with the chloroplast genomes to enable comparative analyses. The RNA editing sites of the five *Deutzia* species were identified and characterized. Combining the chloroplast and transcriptome of five *Deutzia* species, orthologous genes were identified and functionally analyzed. We also identified functional genes involved in monoterpenoid and flavonoid biosynthesis. This study identified interspecific differences and genomic properties of *Deutzia* spp. The results will be helpful for further study of the genomic information and application of the *Deutzia* spp.

## 2. Results

### 2.1. Chloroplast Genomes’ Features

The *Deutzia* chloroplast genome was 156,860–157,025 bp in length, with a typical circular tetrad structure; its GC content was 37.58–37.6% (Appendix A). The chloroplast genomes had highly conserved circular DNA molecules with two inverted repeat (IR) regions that contained IRa and IRb and were separated by a small single copy (SSC) region and a large single copy (LSC) region [34]. We identified 131 genes in five newly measured *Deutzia* spp., including 86 mRNA genes, 37 tRNA genes, and 8 rRNA genes, 16 of which were duplicated in IRs (Figure 1 and Appendix A).

### 2.2. Inverted Repeat Boundary Analysis

During genome evolution, the IR boundaries expanded and contracted, allowing some genes into the IR or SC regions. The chloroplast genome is circular with four boundaries: LSC-IRb, IRb-SSC, SSC-Ira, and IRa-LSC. The IR regions of *Deutzia* spp. were similar in length, from 25,834 to 25,879 bp (Appendix A). Six genes, *rps19*, *rpl2*, *ycf1*, *dnhF*, *tmN*, and *tmH*, were clearly identified in different regions or border regions of the nine chloroplast genomes. Compared with the other species, *D. grandiflora* and *D. hamata* exhibited some significant differences at the left-hand junction (JLB) (IRb/LSC) and right-hand junction (JLA) (IRa/LSC). In both *D. grandiflora* and *D. hamata*, the *rps19* gene was integrated into the IRb region at 24 bp, whereas in the others, it was located exclusively in the LSC region. *Rpl2* in both plants was 79 bp from the JLA/JLB boundary and more than the 43 bp from the boundary in the other species. In addition, the *trnH* gene in *D. glabrata* was 74 bp from the JLA/JLB boundary and was 1–2 bp longer than in the other species.

### 2.3. Comparative Genomic and Pi Analysis

The results of the comparative genomic analysis (Figure 2) showed that *Deutzia* chloroplast genome sequences had a high similarity. No inversions of large segments or gene rearrangements were detected, indicating that the genome structure was highly conserved in terms of gene identity and sequence. Compared with other regions, the IR regions were more conserved. The coding regions were also more conserved than the non-coding regions. In terms of the four components, the SSC region had the highest sequence variation, and the IR regions had the lowest.

The pi values revealed the variation magnitude of variation in the chloroplast genomic sequences of *Deutzia* species, where regions with greater variation provide population genetic and molecular markers. The results showed that *Deutzia* spp. had a low pi value, indicating that the genomic sequence was highly conserved, reaching a maximum of only 0.00625 for coding sequences (Table 1) and 0.01122 for intergenic sequences (Table 2). LSC and SSC regions had higher nucleotide diversity. Thus, the IR regions were more highly conserved than the single-copy regions. The three genes with the highest pi values were *trnL-UAG* (0.00625), *petN* (0.00556), and *rps3* (0.00431) (Table 1). And *trnL-UAG* and *petN* were only 80 and 90 bp in length, respectively. *TrnL-UAG* and *rps3* genes are involved in protein synthesis, and *petN* belongs to the photosynthesis category, which encodes subunits of cytochrome b/f complex. The intergenic regions with the highest pi values were *ndhD*-*psaC*, *petD*-CDS2-*rpoA*, *ccsA*-*ndhD*, and *petB*-CDS2-*petD*-CDS1 (Table 2). These regions may prove useful for molecular markers or further phylogenetic analysis.

### 2.4. Phylogenetic Analysis

To determine the phylogenetic position and evolutionary relationships of *Deutzia* species and other species in the Hydrangeoideae, the coding genes shared in the 37 species were clustered. The phylogenetic tree showed consistency with the traditional morphological classification (Figure 3). *D. grandiflora* and *D. hamata* were clustered with *D. compacta*, and *D. parviflora* was clustered with *D. pilosa*. *D. hypoglauca*, and *D. scabra* were clustered together. All the Hydrangea species were clustered together. Compared with Hydrangea, *Deutzia* species were closer to *Kirengeshoma palmata* and *Philadelphus* species.

### 2.5. KaKs Analysis

Generally, non-synonymous mutations are subject to natural selection pressure, whose effect is indicated by the ratio between KA and KS. Thus, KA/KS > 1 represents a positive selection, and KA/KS < 1 represents a purifying selection. According to the results, the KA/KS values of most genes were less than 1 (Appendix A). It showed that the majority of genes were affected by the purification selection impact. Additionally, the presence of positive selection was demonstrated by *ccsA*, *matK*, *rbcL*, and *ycf1* with KA/KS > 1. In *D. grandiflora*, *D. hamata*, *D. hypoglauca*, and *D. scabra* (Table 3 and Appendix A). *CcsA*, *rbcL*, and *ycf1* are all related to photosynthesis and cellular respiration. These genes may have some effect on the evolution of *D. parviflora* and its adaptation to its environment. The absence of positively selected genes in *D. parviflora* may be related to its relatively poor resistance to stress [35].

### 2.6. Repeat Analysis

#### 2.6.1. Scattered Repetition Analysis

The duplicates of the genus *Deutzia* were similar in type, number, and distribution (Figure 4A,C,E,F). Among the newly measured species, 41, 42, 42, 40, and 45 scattered repeats were found in *D. hamata*, D. *hypoglauca*, *D. scabra*, *D. grandiflora*, and *D. parviflora*, respectively, including 21–24 forward repeats, 18–20 palindromic repeats, and 1–2 reverse repeat/s (*D. polisa* has 2). These repeats were short, most being of only 30–39 bp, with the longest not more than 80 bp. Of the repeats, 47% were located in the CDS, 28% in the intron, and 25% in the intergenic spacer (IGS) (Figure 4D). These repeats could serve as population genetic markers in ongoing studies.

#### 2.6.2. SSR Analysis

The number, type, and distribution of SSRs of *Deutzia* spp. Were similar in both chloroplast and transcriptome. *D. hamata*, *D. hypoglauca*, *D. scabra*, *D. grandiflora*, and *D. parviflora* had 199–201 chloroplast SSRs and 10,685–14,341 transcriptome SSRs. *D. scabra* had significantly fewer single base repeats than other species (Appendix A). More than half of the chloroplast genomes in the genus *Deutzia* were single-base repeats (60–61%), followed by three-base repeats (35%), a small number of two-base repeats (3%), and four-base repeats (1–1.5%). The SSRs distributed in the LSC region were mostly in IGS and CDS regions (Appendix A). The SSRs mainly comprised single nucleotides, (A)n and (T)n, with single base repeats that were mostly 8–21 bp in length. *D. scabra* and *D. hypoglauca* each had an 8-bp C-type SSR.

### 2.7. Chloroplast Genome RNA Editing Analysis

RNA editing is a functionally significant post-transcriptional regulation for chloroplast development in higher plants. Nucleotide changes from cDNA templates may represent real RNA editing sites [36]. By combining transcriptomes, we also identified putative RNA editing sites in the five *Deutzia* species.

Using the PREP-Cp program, 50–60 RNA editing sites were predicted in the five *Deutzia* chloroplast genomes, and 29–31 editing events were identified through RNA-Seq mapping. All editing events occurred in association with C-to-U substitutions, and the *ndh*, *rpo*, *rps*, and *atp* genes had the most editing sites (Figure 5). This result suggested that RNA editing may be related to the synthesis and function of subunits of NADH dehydrogenase, RNA polymerase, ATP synthase, and proteins in small ribosomal subunits. Serine to leucine accounted for the largest ratio of amino acid changes resulting from these RNA edits (Appendix A), and most of the substitutions occurred at the second codon position.

### 2.8. Orthologous Gene Analysis of Transcriptome and Chloroplast Genome

In the process of organelle formation, most ancestral genes are transferred to the host nuclear genome and integrated into the nuclear region. Extensive DNA sequence transfer from chloroplast to nuclear genomes remains a common phenomenon in plants. Using OrthoFinder software, 50–56 orthologous gene groups and 163–194 orthologous genes were identified, and the protein sequences were extracted from the chloroplast genomes and transcriptomes of five *Deutzia* species.

The sequences of orthologous genes were annotated using the COG database (Figure 6). The main functions of the orthologous genes included energy production and conversion, translation, ribosomal formation, general function prediction, post-translational modification, and protein transport. Some were annotated into two functional categories. To fully reflect the functional categories of the orthologous genes, they were also subjected to GO annotation (Figure 7) to determine the biological processes, cellular components, and molecular functions related to the orthologous genes.

The proteins of the orthologous gene were located in different cellular components, including chloroplast, mitochondria, nucleus, and cytoplasm (Figure 8). Some proteins were annotated to two or three cellular components at the same time. For example, TRINITY_DN2475_c0_g1 belonged to the mitochondrion and chloroplast thylakoid membrane of *D. hamata* encoding ATP-dependent Clp protease proteolytic subunit 2 (Appendix A). In *D. hamata*, *D. hypoglauca*, *D. parviflora*, and *D. scabra*, 1 protein was annotated to mitochondria, chloroplast, and small ribosomal subunits, respectively (Appendix A), and 3–5 proteins were annotated to both the ribosome and chloroplast in *D. hamata*, *D. hypoglauca*, *D. parviflora*, and *D. scabra*. The main functions of these proteins were related to translation, ribosomal structure, and biogenesis. Some proteins were annotated in both the nucleus and cytoplasm; these are related to the intracellular membrane-bounded and perinuclear region of the cytoplasm (Appendix A).

The RT-qPCR result showed that the expression levels of most orthologous genes were lower than chloroplast expression, with the exception of *rpl14*-B of *D. hamata*, and most chloroplast genes had higher expression levels except *rpl14* and *rps11* of *D. parviflora* (Figure 9). And in *D. grandiflora*, *rps11* and *rps19* had higher expression levels in the chloroplast, and *rpl14* and *ycf3* of *D. scabra* were also higher in the chloroplast.

All pathways involved in orthologous genes were analyzed using the KEGG database (Appendix A). Orthologous genes were mainly associated with pathways, such as ribosomes, metabolic pathways, and oxidative phosphorylation, suggesting that these orthologous genes may have an impact on plant growth and development. In addition, some homologous genes were also associated with chemical carcinogenesis-reactive oxygen species (Figure 10), suggesting that the genus *Deutzia* may have chemical toxicity. All of these associations need to be investigated in depth and verified.

### 2.9. Discovery of Genes Potentially Encoding Medicinal Ingredients

Monoterpenoid is a type of terpenoid that is typically volatile and highly aromatic; some also have antioxidant, antibacterial, and anti-inflammatory activities, among other physiological properties. Monoterpenoids are important raw materials in the pharmaceuticals, foods, and cosmetic industries. Flavonoids and iridoids in the leaves of *Deutzia* spp. [37,38,39] are active chemical constituents in all species. KEGG pathway analysis revealed 11–17 genes annotated to the monoterpenoid biosynthesis pathway in the five species of *Deutzia*, which were mainly related to the biosynthesis of indole alkaloid, linalool synthase, neomenthol, and 4,5-dihydro-5,5-dimethyl-4-(3-oxobutyl) funan-2(3H)-1-hexanoate (Appendix A). There were 32–49 genes annotated to the flavonoid biosynthesis pathway; these were involved in the production of numerous flavonoid compounds, such as gallocatechin and pinobanksin3-acetate (Appendix A). This finding suggests that *Deutzia* plants have potential as pharmaceutical or chemical raw materials.

## 3. Discussion

### 3.1. Relatively Conserved Chloroplast Genome of Deutzia Plants

The structure and size of the *Deutzia* spp. chloroplast genome, which had a typical double-stranded circular tetrad structure, was highly conserved in this study. The sequence length was 156,860–157,025 bp, the GC content was 37.58–37.6%, and 131 genes were encoded. The IR regions of *Deutzia* species were similar in length from 25,834–25,879 bp. The chloroplast genome sequences were highly similar, and no large fragment inversion or gene rearrangement was detected, indicating that the genome had a highly conserved structure according to the gene identity and order.

The pi values of *Deutzia* were very low, with the highest being only 0.00625 (*trnL-UAG*), indicating that the nucleic acid sequence variation was low, and the genome sequence was highly conserved. In some species, the *trnL-UAG* region also showed higher sequence differences [40]. In general, tRNA genes are always highly conserved, and the intergenic region may have a relatively faster evolutionary rate [41,42]; for example, *rpl32-trnL*^(UAG)^ was highly variable in many species [42,43,44]. Ka/Ks analysis showed that *ccsA*, *matK*, *rbcL*, and *ycf1* were affected by positive selection in *D. grandiflora*, *D. hamata*, *D. hypoglauca*, and *D. scabra*, indicating that these genes may be related to environmental adaptation during the evolution of *Deutzia* plants. No positive selection event was found in *D. parviflora*, which may explain its relatively poor stress resistance. In *Camellia*, *ycf1* was affected by positive selection [45]. *CcsA*, *matK*, *rbcL*, and *ycf1* were also found as potential mutational hotspot genes [45,46,47,48,49], which could be used for phylogenetic analysis or as a plastid barcode.

*Deutzia* spp. had conserved scattered repeats and SSRs. The repeat sequences of *Deutzia* were similar in number and distribution, more than half of which were mononucleotide repeats (60–61%), followed by trinucleotide repeats (35%), small proportions of dinucleotide repeats (3%), and tetranucleotide repeats (1–1.5%).

### 3.2. Evolution Analysis

Previous phylogenetic studies of the Hydrangeae support the position of *Deutzia* in Philadelphiae [50,51]. The shared protein-coding genes of chloroplast genome sequences of 32 Hydrangeae species downloaded from NCBI, and 5 *Deutzia* species were clustered, analyzed, and constructed a phylogenetic tree. *D. grandiflora* and *D. hamata* were clustered together in a subsection with *Grandiflorae* [3]. *Deutzia parviflora* was clustered with *D. Pilosa*, and *D. glabrata* was clustered with *D. crassifolia.* In terms of morphological classification, *D. parviflora*, and *D. crassifolia* belonged together in Ser. *parviflorae*. All of the Hydrangea species were clustered together.

In a previous study, ribosomal DNA ITS and 26S and chloroplast regions matK, rbcL, and trnL-F, as well as morphological characteristics, were used for phylogenetic analysis [9]. In that study, *D. compacta*, *D. crassiflora* and *D. pilosa* belonged to subclade a, with weak bootstrap support; the geographic distribution was thought to be related. *D. scabra* was classified in subclade b, with strong support. *D. glabrata* and *D. parviflora* belonged to subclade c with not very strong support (BP = 76); they had similar appearance characteristics. *D. grandiflora* and *D. hypoglauca* belonged to subclade d. [9] These previous results may have differed from ours because the shared genes used for the analyses were different. Additionally, in our results, species of Sect. *Mesodeutzia* and Sect. *Deutzia* were all mixed together; thus, we consider the proposed merging of Sect. *Mesodeutzia* and *Deutzia* to be reasonable. However, due to limitations in the material collection process, we did not assess additional species of the genus *Deutzia*, and it is striking that genotypes of plastid are often associated by geographical location rather than taxonomic relationships [52]. As such, the evolution of *Deutzia* plants still needs to be explored more extensively at the molecular level.

Based on morphological and molecular evidence, *Deutzia* is closely related to *Kirengeshoma* and *Philadelphus* closely [1,50,51,53,54]. In the *Flora Reipublicae Popularis Sinicae*, the genus *Deutzia* is classified into the Tribe Philadelpeae Reich., but the evolutionary tree showed that *Deutzia* was closer to *K. palmata* of the Tribe Kirengeshomeae Engl. This result is also consistent with previous studies [55].

### 3.3. Functional Annotation of Chloroplast and Transcriptome Orthologous Genes and Genes Related to Major Active Ingredients

In plants, there are three major genetic systems: chloroplasts, mitochondria, and the nucleus, in which genes from different systems interact to regulate growth and development. During the course of evolution, there have been frequent exchanges of genes among the different genetic systems. DNA movement occurs when genes are transferred from organelles to the nucleus [19,56]. In this study, we identified 163–194 orthologous genes of chloroplasts and transcriptomes of five species of *Deutzia* spp. These included orthologous genes of chloroplasts, mitochondria, and nuclear genes; 19–23 genes were annotated to the mitochondrion and 7–9 to the nucleus. The qPCR of genes annotated to the nucleus revealed that these genes had various expression degrees. This result indicated that there was some gene exchange between chloroplast and mitochondria, chloroplast, and the nucleus. The functions of orthologous genes were mainly related to energy metabolism, translational and post-translational modifications, and protein transport. The pathways involved in the orthologous genes were mainly related to the ribosome, metabolic, and oxidative phosphorylation pathways, suggesting that these orthologous genes may be related to the physical growth of plants. There were also genes annotated to chemical carcinogenesis.

### 3.4. Genes Related to Major Active Ingredients

The chemical constituents of the traditional Chinese medicine developed from *Deutzia* are mainly flavonoids and iridoids. Analysis of transcriptomic data on the KEGG pathway showed that 11–17 genes in five *Deutzia* spp. were annotated to the monoterpenoid biosynthesis pathway, which was mainly associated with the biosynthesis of indole alkaloid, linalool synthase, neomenthol, and 4,5-dihydro-5,5-dimethyl-4-(3-oxobutyl)funan-2(3H)-1-hexanoate biosynthesis. Notably, 32–49 genes were annotated to the flavonoid biosynthesis pathway, which is involved in the production of numerous flavonoids.

In Chinese medical texts, urination clears heat and diuresis and is often applied externally or in a decoction for external washing [57,58]. However, *Deutzia* is poisonous and should be used with caution [4]. Analysis of the chloroplast and transcriptome orthologous genes and of genes related to the herbal components obtained from the transcriptome confirmed this and, to some extent, explained the medicinal use of *Deutzia* spp. However, the utilization of *Deutzia* spp. in modern medicine is relatively rare, and considering its toxicity, further trials and studies are needed before it can be widely utilized.

## 4. Materials and Methods

### 4.1. Plant Material and Chloroplast Genome Sequences Collection

Fresh leaves of five wild-type *Deutzia* species, *D. hamata*, *D. hypoglauca*, *D. scabra*, *D. grandiflora*, and *D. parviflora*, growing in an outdoor natural climate, were used in this experiment. *Deutzia hypoglauca* and *D. scabra* were obtained from Hebei Agricultural University (115.44° E, 38.83° N, temperate continental monsoon climate), and *D. grandiflora* and *D. parviflora* were collected from Shijiazhuang Botanical Garden (114.38° E, 38.11° N, temperate continental monsoon climate). Three plants of each *Deutzia* spp. (*D. grandiflora*, *D. hamata*, *D. hypoglauca*, *D. parviflora*, and *D. scabra*) with the same health and growth status were selected as replicates. After quick freezing in liquid nitrogen, DNA and RNA were extracted then sent to Genepioneer Biotechnologies (Nanjing, Jiangsu, China) for sequencing. Complete chloroplast genome sequences of 32 other plants (Table 4) that were collected from the NCBI were used for comparison with those obtained by sequencing.

### 4.2. Chloroplast Genome Sequencing, Assembly, and Annotation of Five Deutzia Species

Fresh leaves of three plants of each *Deutzia* spp. (*D. grandiflora*, *D. hamata*, *D. hypoglauca*, *D. parviflora*, and *D. scabra*) with the same health and growth status were first quickly frozen using liquid nitrogen and then used to extract total genomic DNA via a Plant Genomic DNA Kit (Tiangen, Beijing, China). The genomic DNA was sequenced on Illumina Novaseq 6000 platform (San Diego, CA, USA). Raw reads were filtered using fastp software (version 0.20.0; https://github.com/OpenGene/fastp, accessed on 8 July 2022) to obtain high-quality reads (i.e., clean data).

SPAdes v3.10.1 software [59] (http://cab.spbu.ru/software/spades/, accessed on 8 July 2022) was used to assemble the core module and chloroplast genome. Chloroplast coding sequences (CDSs) were annotated using Prodigal v2.6.3 (https://www.github.com/hyattpd/Prodigal, accessed on 8 July 2022). Hmmer v3.1b2 (http://www.hmmer.org/, accessed on 8 July 2022) was used to predict rRNA. Aragorn v1.2.38 (http://130.235.244.92/ARAGORN/, accessed on 8 July 2022) was used to predict tRNA. The reference sequence NC_044843.1 was used for quality control after assembly. The assembled sequences were annotated a second time with Blast v2.6 (https://blast.ncbi.nlm.nih.gov/Blast.cgi, accessed on 8 July 2022). Five *Deutzia* chloroplast genomes have been uploaded to GenBank. The GenBank accession numbers are as follows: *Deutzia hamata* OR415702, *Deutzia hypoglauca* OR415703, *Deutzia parviflora* OR415704, *Deutzia grandiflora* OR415705, *Deutzia_scabra* OR415706. Circular and linear chloroplast genome maps were drawn using OGDRAW online software (https://chlorobox.mpimp-golm.mpg.de/OGDraw.html, accessed on 8 July 2022) and Circos v0.69-6 software [60].

### 4.3. Transcriptome Sequencing, Assembly, and Annotation

Three plants of each *Deutzia* spp. (*D. grandiflora*, *D. hamata*, *D. hypoglauca*, *D. parviflora*, and *D. scabra*) with the same health and growth status were selected as replicates. Fresh leaves were first quickly frozen using liquid nitrogen and then used to extract total RNA using the TRIzol method (Invitrogen, Carlsbad, CA, USA) from fresh leaves. The Illumina Novaseq 6000 platform (San Diego, CA, USA) was used for sequencing. The raw reads were filtered using FastQc (v0.11.8) software [61] to obtain high-quality clean data.

Transcriptome was assembled using Trinity [62]. Gene function was annotated based on the Nr (NCBI non-redundant protein sequences), Nt (NCBI non-redundant nucleotide sequences), Pfam (Protein family), KOG/COG (Clusters of Orthologous Groups of proteins), Swiss-Prot, and KEGG ortholog (KO) databases.

### 4.4. Analysis of Repeat Structures of Transcriptome and Chloroplast Genome

MISA v1.0 software (http://pgrc.ipk-gatersleben.de/misa/misa.html) was used to analyze the SSR structures, with parameters 1–8 (≥8 single base repeats), 2–5, 3–3, and 4–3 selected. Duplicate sequences were identified using vmatch v2.3.0 software (http://www.vmatch.de/), with the following settings: 30 bp minimum length, hamming distance = 3, and identification = forward, palindromic, reverse, and complement.

### 4.5. KaKs Analysis

Gene sequences were aligned using the mafft v7.310 software (https://mafft.cbrc.jp/alignment/software/, accessed on 8 July 2022) and KaKs_Calculator v2.0 software (https://sourceforge.net/projects/kakscalculator2/, accessed on 8 July 2022) was used to calculate gene Ka (synonymous) and /Ks (nonsynonymous) values, following the MLWL method.

### 4.6. Chloroplast Genomes Comparative Analysis

Genome global alignment was performed using MAFFT v7.310 software (https://mafft.cbrc.jp/alignment/software/, --auto mode) for SNP and INDEL identification (mpileup -m 2 -F 0.002 -d 1000). Global alignment of homologous gene sequences of *Deutzia* spp. was performed using MAFFT software (--auto mode). Nucleic acid diversity (pi) values of genes were calculated using DNAsp5. A comparative analysis of the chloroplast genome structure was performed using CGVIEW software (http://stothard.afns.ualberta.ca/cgview_server/, accessed on 8 July 2022). Perl-IRscope (https://github.com/xul962464/ perl-IRscope, accessed on 8 July 2022) was used to visualize the boundaries. Genomic comparisons were performed and visualized using Mauve software (http://darlinglab.org/mauve, accessed on 8 July 2022).

### 4.7. Evolution

Whole genomes of the 37 species were subjected to evolutionary tree analysis. All loop sequences had the same starting point, and sequences were compared between species using MAFFT software (v7.427, --auto mode). The resulting data were compared using RAxML v8.2.10 software (https://cme.h-its.org/exelixis/software.html, accessed on 8 July 2022), and a maximum likelihood evolutionary tree was constructed.

### 4.8. RNA Editing Site Identification

The PREP-Cp online tool [63] was used to identify chloroplast RNA editing sites of five *Deutzia* species (cut-off value = 0.8), and the results were combined with those of RNA-Seq read mapping. TopHat v2.1.0 [64] was used to map RNA-Seq reads from *Deutzia* spp. to the chloroplast genome. SAMtools [65] was used to combine SNPs and identify editing sites in chloroplast genomes. The mapped reads were visualized using IGV (Integrative Genomics Viewer) [66]. RNA editing efficiency was equal to the number of edited reads divided by the number of mapped reads [67].

### 4.9. Identification and Analysis of Orthologous Genes of Chloroplast Genome and Transcriptome

Using OrthoFinder software [68,69] and BLAST [70] with the default settings, we identified homologous gene groups in the chloroplast genome and transcriptome, extracted protein sequences, and performed COG, Gene Ontology (GO), and Kyoto Encyclopedia of Genes and Genomes (KEGG) annotations.

Orthologous genes of the chloroplast and nucleus were verified using a qPCR test. The primers were designed by Primer Premier 6 [71] and synthesized by Tsingke Biotechnology (Beijing, China); primer information is provided in Appendix A. Chloroplast of *Deutzia* spp. was separated by a Minute Chloroplast Isolation Kit (Invent, Beijing, China), chloroplast RNA was extracted using Plant Total RNA Isolation Kit Plus (Foregene, Chengdu, Sichuan, China), and chloroplast RNA and total RNA were reverse-transcribed using a PrimeScript II 1st Strand cDNA Synthesis Kit (TAKARA, Beijing, China). MagicSYBR Mixture (CWBIO, Beijing, China) and Agilent Mx3005P (Stratagene, Palo Alto, CA, USA) were used to conduct the qPCR test.

### 4.10. Discovery of Genes Potentially Encoding Medicinal Ingredients

Combined with the KEGG results, we analyzed the function of genes in the monoterpenoid biosynthesis and flavonoid biosynthesis pathways from transcriptome data.

## 5. Conclusions

To investigate the phylogeny of *Deutzia* spp., RNA editing of the chloroplast genome, and exchange between chloroplast and nuclear genes, we sequenced, assembled, annotated, and analyzed the chloroplast genomes and transcriptomes of five species of *Deutzia* spp. (*D. hamata*, *D. hypoglauca*, *D. scabra*, *D. grandiflora*, and *D. parviflora*). The chloroplast genomes of *Deutzia* spp. were typical circular genomes 156,860–157,025 bp in length, encoding 131 genes, and the genome structures were conservative and stable. Phylogenetic analyses demonstrated that *Deutzia* spp. were closely related to *Kirengeshoma* and *Philadelphus*. Through chloroplast genomic and transcriptomic analysis, 29–31 RNA editing sites were identified in five *Deutzia* species. The *ndh*, *rpo*, *rps*, and *atp* genes had the most abundant editing sites, and 163–194 orthologous genes associated with ribosome, metabolic, and oxidative phosphorylation pathways were identified.

## Figures and Tables

**Figure 1 ijms-24-12954-f001:**
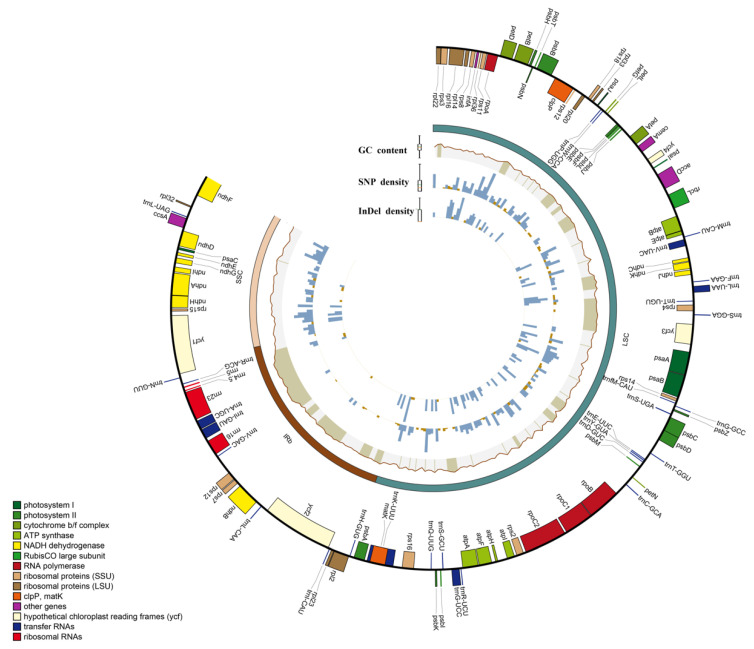
Circos diagram of the basic features of the five genomes. Rectangles at the outermost are the chloroplast genes of different functional groups with color coding.

**Figure 2 ijms-24-12954-f002:**
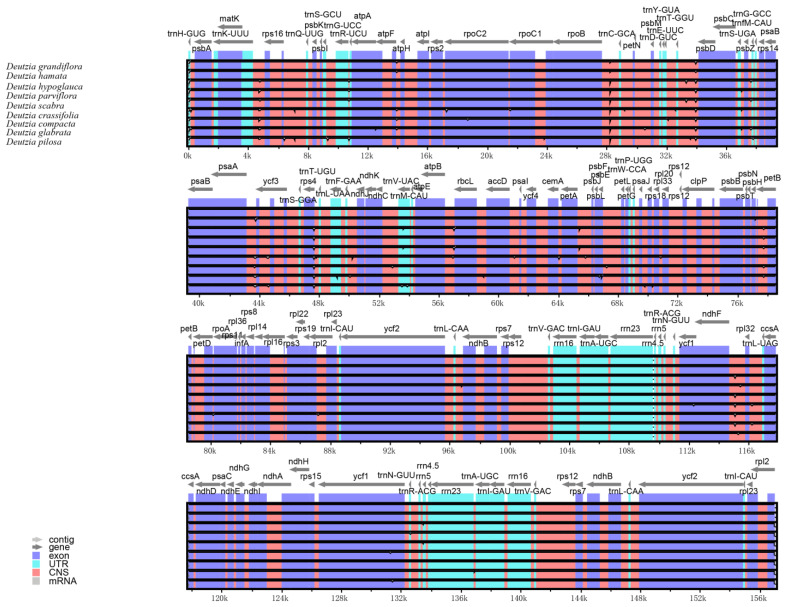
Comparative genomic analysis among nine *Deutzia* chloroplast genome sequences.

**Figure 3 ijms-24-12954-f003:**
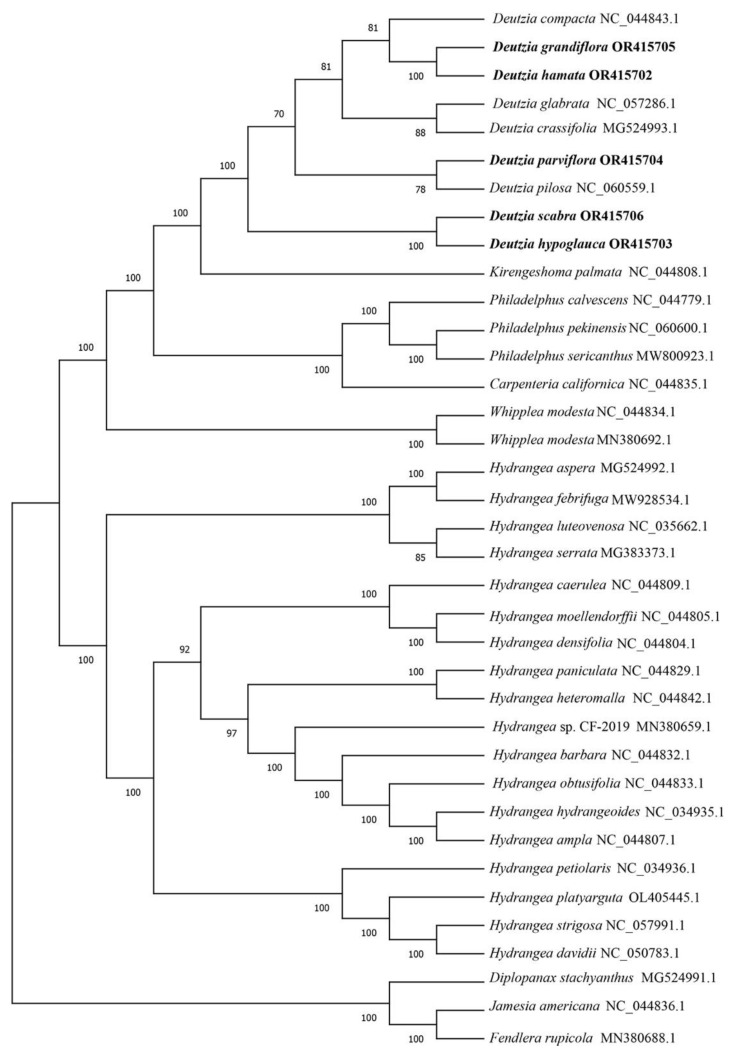
Phylogenetic tree of 37 species. The number on the branch represents reliability. The larger the number, the higher the credibility, up to 100.

**Figure 4 ijms-24-12954-f004:**
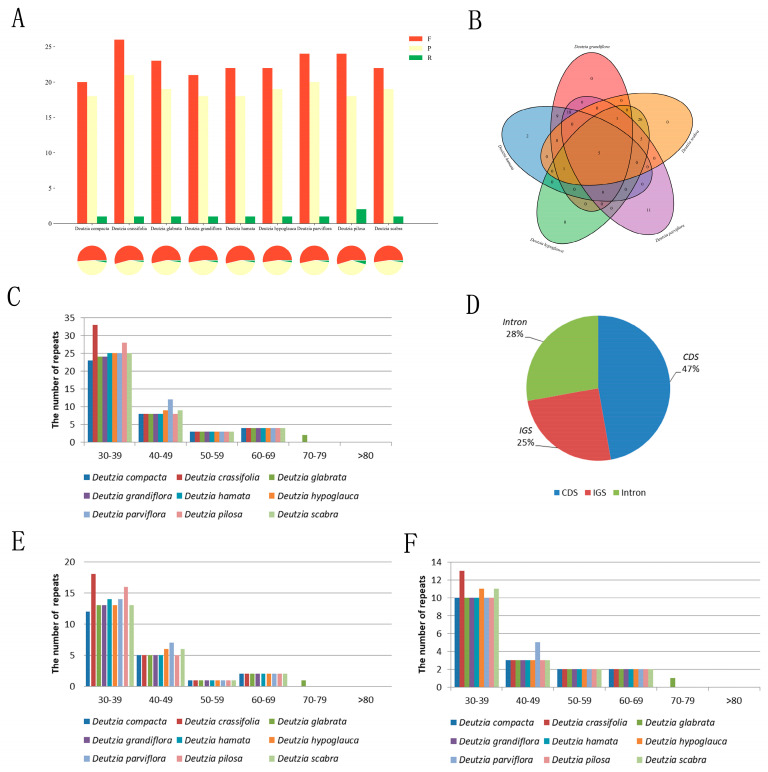
Scattered repeats sequence statistics of Deutzia. (**A**): Results of scattered repeats sequence statistics; (**B**): Shared repeats of five *Deutzia* species; (**C**): Distribution of scattered repeats; (**D**): Location of scattered repeats; (**E**): Frequency of forward repeats; (**F**): Frequency of palindromic repeats.

**Figure 5 ijms-24-12954-f005:**
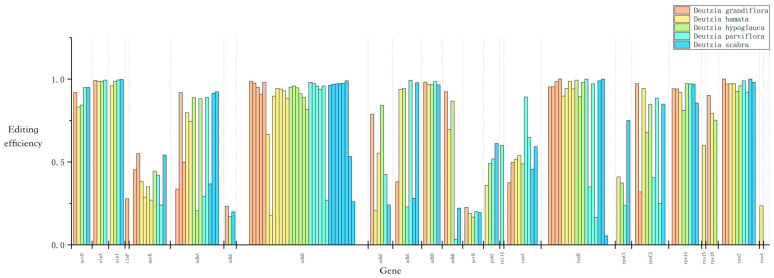
Chloroplast RNA editing of *Deutzia* species.

**Figure 6 ijms-24-12954-f006:**
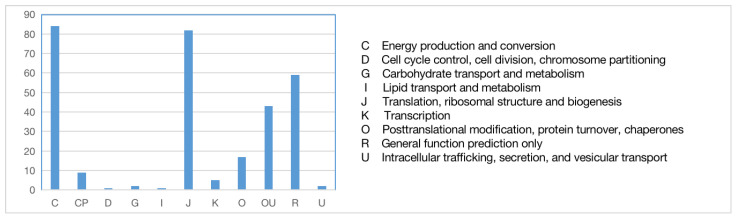
COG functional annotation of orthologous genes.

**Figure 7 ijms-24-12954-f007:**
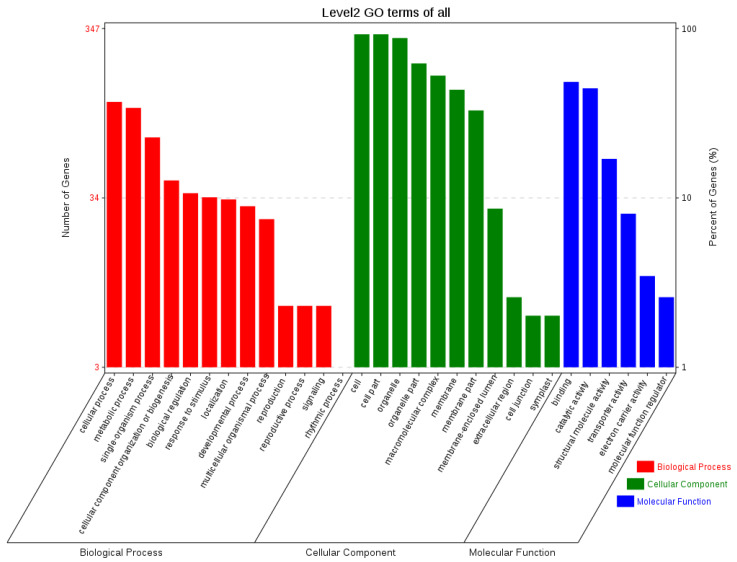
GO functional annotation of orthologous genes.

**Figure 8 ijms-24-12954-f008:**
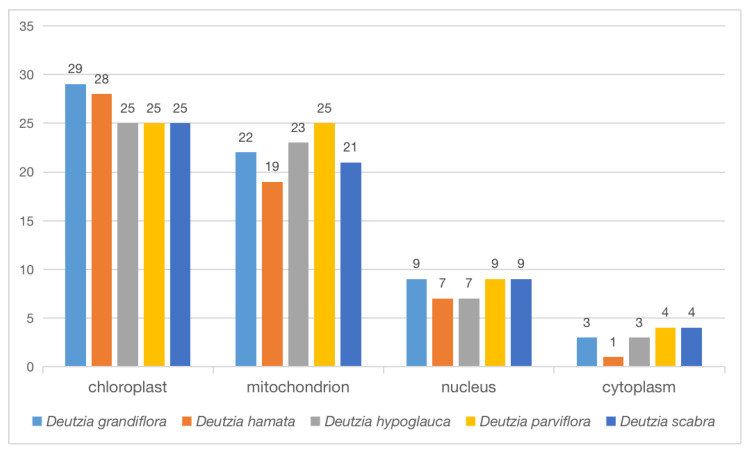
Cellular components of orthologous proteins.

**Figure 9 ijms-24-12954-f009:**
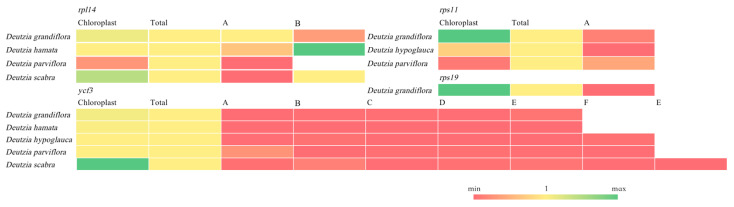
RT-qPCR of chloroplast RNA, total RNA, and orthologous genes.

**Figure 10 ijms-24-12954-f010:**
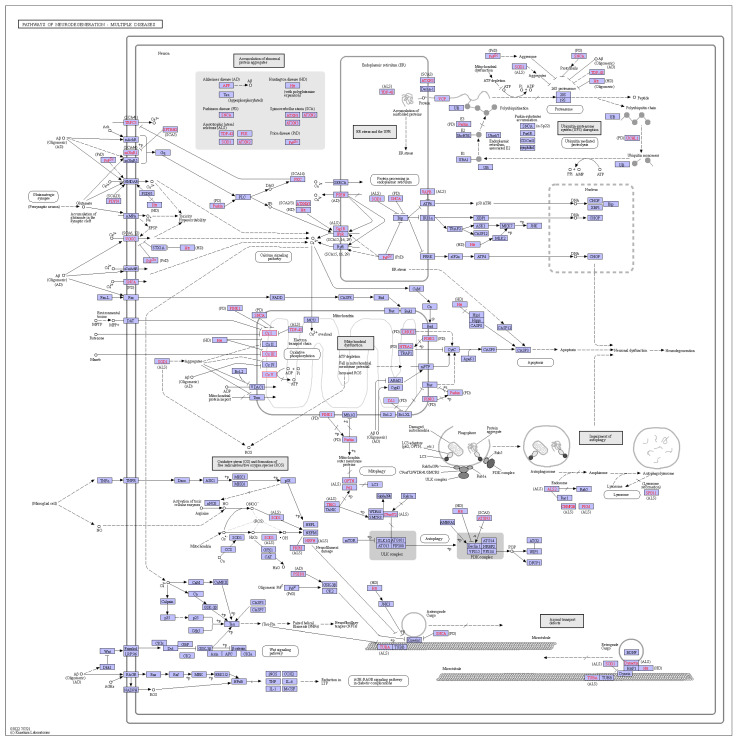
Chemical carcinogenesis—reactive oxygen species pathway.

**Table 1 ijms-24-12954-t001:** Pi value of the coding sequences.

Region	Pi	Total Number of Mutations	Region Length (bp)
SSC.gene4.*trnL-UAG*	0.00625	1	80
LSC.gene21.*petN*	0.00556	1	90
LSC.gene82.*rps3*	0.00431	5	657
LSC.gene53.*cemA*	0.00266	3	690
LSC.gene17.*rpoC2*	0.00253	20	4158
LSC.gene5.*rps16*	0.00253	1	264
SSC.gene5.*ccsA*	0.00242	4	966
LSC.gene78.*infA*	0.00214	1	234
LSC.gene49.*rbcL*	0.0021	5	1428
LSC.gene4.*matK*	0.00209	6	1518

**Table 2 ijms-24-12954-t002:** Pi value of the intergenic sequences. CDS1 and CDS2 were the different exons of the same gene.

Region	Pi	Total Number of Mutations	Region Length (bp)
SSC.gene4.*ndhD*-*psaC*	0.01122	3	132
LSC.gene59.*petD*-CDS2-*rpoA*	0.01099	5	187
SSC.gene3.*ccsA*-*ndhD*	0.00889	6	258
LSC.gene57.*petB*-CDS2-*petD*-CDS1	0.00802	7	196
LSC.gene10.*atpF*-CDS1-*atpH*	0.00734	8	355
LSC.gene13.*rps2*-*rpoC2*	0.00704	5	225
SSC.gene2.*rpl32*-*ccsA*	0.00669	21	1148
LSC.gene14.*rpoC2*-*rpoC1*-CDS2	0.00654	5	174
LSC.gene46.*rps18*-*rpl20*	0.00591	4	298
SSC.gene4.*ndhD*-*psaC*	0.01122	3	132

**Table 3 ijms-24-12954-t003:** Genes subjected to purification selection effect in *Deutzia* species. Blank means there is no purification selection here.

	*Deutzia grandiflora*	*Deutzia hamata*	*Deutzia hypoglauca*	*Deutzia scabra*
	vs	vs	vs	vs
*ccsA*	*Deutzia pilosa*	*Deutzia pilosa*		
*matK*	*Deutzia glabrata*	*Deutzia glabrata*	*Deutzia glabrata*	*Deutzia glabrata*
*rbcL*	*Deutzia pilosa*	*Deutzia pilosa*	*Deutzia pilosa*	*Deutzia pilosa*
*ycf1*			*Deutzia glabrata*	*Deutzia crassifolia* *Deutzia glabrata*

**Table 4 ijms-24-12954-t004:** Species and their NCBI accession numbers.

Number	Species	Accession	Number	Species	Accession
1	*Deutzia glabrata*	NC_057286.1	17	*Hydrangea caerulea*	NC_044809.1
2	*Deutzia crassifolia*	MG524993.1	18	*Hydrangea serrata*	MG383373.1
3	*Deutzia pilosa*	NC_060559.1	19	*Hydrangea barbara*	NC_044832.1
4	*Deutzia compacta*	NC_044843.1	20	*Hydrangea petiolaris*	NC_034936.1
5	*Philadelphus calvescens*	NC_044779.1	21	*Hydrangea moellendorffii*	NC_044805.1
6	*Philadelphus sericanthus*	MW800923.1	22	*Hydrangea febrifuga*	MW928534.1
7	*Philadelphus pekinensis*	NC_060600.1	23	*Hydrangea aspera*	MG524992.1
8	*Kirengeshoma palmata*	NC_044808.1	24	*Hydrangea luteovenosa*	NC_035662.1
9	*Carpenteria californica*	NC_044835.1	25	*Hydrangea obtusifolia*	NC_044833.1
10	*Whipplea modesta*	MN380692.1	26	*Hydrangea hydrangeoides*	NC_034935.1
11	*Hydrangea davidii*	NC_050783.1	27	*Hydrangea ampla*	NC_044807.1
12	*Hydrangea sp.* CF-2019	MN380659.1	28	*Hydrangea densifolia*	NC_044804.1
13	*Hydrangea strigosa*	NC_057991.1	29	*Jamesia americana*	NC_044836.1
14	*Hydrangea heteromalla*	NC_044842.1	30	*Diplopanax stachyanthus*	MG524991.1
15	*Hydrangea paniculata*	NC_044829.1	31	*Whipplea modesta*	NC_044834.1
16	*Hydrangea platyarguta*	OL405445.1	32	*Fendlera rupicola*	MN380688.1

## Data Availability

The original contributions presented in the study are not publicly available at present.

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
