# Peer review of "Analysis of the RNA Editing Sites and Orthologous Gene Function of Transcriptome and Chloroplast Genomes in the Evolution of Five Deutzia Species"

_ijms, 2023, doi:10.3390/ijms241612954_

Round 1
Reviewer 1 Report
Please see accompanying file.

Author Response
Response to Reviewer 1 Comments
Dear reviewer:
Thank you for your comments on our manuscript entitled “Analysis of the RNA Editing Sites and Orthologous Genes Function of Transcriptome and Chloroplast Genome and the Evolution of Five Deutzia Species” (ID: IJMS-2249135). We appreciate the time and effort that you have invested in providing us with valuable comments and suggestions for improving our work. We have carefully studied all the comments and suggestions and made the necessary revisions to improve the clarity and quality of our work which we hope meet with approval. And the revised version was marked by “Track Changes” function. The main corrections in the paper and responds to the comments are as following:
Responds to the reviewer’s comments:
Point 1: There are problems of the types of RNA editing sites, editing efficiency and reliability. the discussion of non-C-U editing sites is too speculative which should be deleted.
Response 1: Considering the Reviewer’s suggestion, we have reanalyzed the data. In this experiment, we first used the chloroplast genome to predict RNA editing sites, then mapped the transcriptome and predicted results, and finally obtained real editing sites. In our previous results, due to the lack of appropriate screening conditions, some sites that may not be accurately edited were also included, such as sites with low editing efficiency. And we did identify that atpF had two mutations, but it was omitted due to the low editing efficiency and few edition events in DNA depth during the screening process. After re-screening, we obtained and re-analyzed the sites with relatively high editing efficiency, and found that there were only 1-3 non-C-to-U editing sites in each Deutzia species, which should be relatively reliable. The third site of each species atpF contains a U-to-G with relatively low editing efficiency (39%-63%). The table has been updated, and the data of other species are in the back tab. And we have deleted the too speculative paragraphs in the discussion.
Point 2: The expression of the transcriptome type used is unclear. The cell compartments where the homologous genes were located is not expressed. Some language expressions are unclear, and the part of the homologous gene function should be deleted.
Response 2: We are very sorry for the errors in our writing. We have modified the ambiguous sentences and deleted the over-speculated paragraphs.
The transcriptome we use is a usual transcriptome that includes all the transcriptional information of the leaves. In the previous version, we really didn 't think carefully enough. By GO annotation of orthologous genes, we obtained the cell compartments where these genes were located.
Genes for monoterpene biosynthesis and flavonoid biosynthesis were derived from transcriptomes rather than homologous genes. And some orthologous genes were annotated to two pathways at the same time.
Point 3: Tables and pictures on nucleic acid diversity need to be improved. Three genes with the highest pi value need to be discussed, especially trnL-UAG.
Response 3: We are very sorry for our negligence of these problems. We have updated the pictures and tables and discussed these genes. The tRNA genes are generally conserved, although trnL-UAG has the highest pi value of 0.00625, this value generally appears to be relatively low (combined with the results of other species). Rapid evolution of petN and rps3 were rarely found in other flowering
plants.
Point 4: The specific value of KAKS is lacking. The expression of positive selection is not accurate enough.
Response 4: We are very sorry for our negligence of these problems. We have added supplementary tables with detailed data. CcsA, matK, rbcL, and ycf1 genes were mostly used as hotspots in other flowering plants for molecular markers. We have supplemented the discussion of these genes and modified the inaccurate description.
Point 5: Phylogenetic analysis needs to be discussed in combination with previous results.
Response 5: Considering the Reviewer’s suggestion, we have improved the discussion of phylogenetic analysis combining the results of previous studies.
Point 6: Extensive corrections to the English are needed (grammar, typos, word usage etc.).
Response 6: We are very sorry for the errors in our writing. We have carefully gone over and revised the grammar of the full text and corrected the miscellaneous and inappropriate vocabulary.

Reviewer 2 Report
To,
The Editor,
IJMS, MDPI,
Manuscript ID: ijms-2249135
Subject: Submission of comments on the manuscript in “IJMS"
Dear Editor IJMS, MDPI,
Thank you very much for the invitation to consider a potential reviewer for the manuscript (ID: ijms-2249135). My comments responses are furnished below as per each reviewer’s comments.
Dear Chief Editor,
The present work authors, the chloroplast genomes and transcriptomes of five Deutzia genus species were characterized and made combine analysis and a phylogenetic tree was constructed with 32 other chloroplast genome sequences of Hydrangeoideae species. The results showed that the five Deutzia chloroplast genomes had typical double-stranded circular tetrad structures with 156,860−157,025 bp genome sizes, and the guanine-cytosine (GC) content was 37.58−37.6%. The gene structure, type, and number were similar, sequence variation was small, and the gene structures were stable. The Deutzia species were closely related to Kirengeshoma palmata and Philadelphus. By combining chloroplast genomic and transcriptomic analyses, 34−39 RNA editing events and types other than C-to-U substitutions were identified in the five Deutzia species and 163−194 orthologous genes were identified, which may be important for the growth and development, as well as medicinal functions, of Deutzia spp. The manuscript represents a very important piece of research in a logical presentation. Therefore, it might be conditionally accepted as subject to major revision. Instead, authors have to improve their manuscripts with many non-clear meanings, inaccuracies, and the authors need to address the following issues before it can be accepted for publication.
- I have read the entire manuscript and my initial comment is that manuscript is poorly written. I have significant concerns about the grammar and vocabulary of the manuscript; therefore, I recommend the authors to used an English proofreading service..
- The structure of the abstract should be improved, as well as the lack of several aspects that should be included in this section. Most of the abstracts contain confusing and uninformative sentences. Please give more precise objectives here (such as in the Abstract). The abstract should highlight the most important results of the parameters and characteristics assayed.
- Keyword must in alphabtical order.
- Introduction grammatical issues appear to be most prevalent in the introduction, making for very confusing reading. Further, the introduction is short but has no clear thread.
- Why you selected this crop for your experiment? Please provide the detail of the used variety.
- The figures are quite low resolution and difficult to make out. Higher-resolution versions will be needed for publication. Further, text in figure is not readble. for example, in Figures 1, 2, 3, 4, 5, 6, 7, 8, 9 and 10.
- In Material and Methods:- indicate how many replicates assayed in each analysis/parameter. The number of samples or biological and technical replicates should be mentioned for each parameter in the methods.
- The discussion should be interpreted with the results as well as discussed in relation to the present literature.
- Conclusion section is very lengthy. The author should emphasize this in a better way.
- References: shall have to correct the whole References according to the ”Instructions for the Authors”, e.g. title should not be in italics, the Journal name is in italics, and the author shall have to use the abbreviated name Journals cited the year must be bold, the scientific name must be italics etc. Please check all references carefully.
Author Response
Response to Reviewer 2 Comments
Dear reviewer:
Thank you for your comments on our manuscript entitled “Analysis of the RNA Editing Sites and Orthologous Genes Function of Transcriptome and Chloroplast Genome and the Evolution of Five Deutzia Species” (ID: IJMS-2249135). We appreciate the time and effort that you have invested in providing us with valuable comments and suggestions for improving our work. We have carefully studied all the comments and suggestions and made the necessary revisions to improve the clarity and quality of our work which we hope meet with approval. And the revised version was marked by “Track Changes” function. The main corrections in the paper and responds to the comments are as following:
Responds to the reviewer’s comments:
Point 1: There are grammar and vocabulary problems in the manuscript.
Response 1: We are very sorry for the errors in our writing. We have carefully gone over and revised the grammar of the full text and corrected the miscellaneous and inappropriate vocabulary.
Point 2: The structure of the abstract should be improved.
Response 2: Considering the Reviewer’s suggestion, we have adjusted the structure of the abstract, deleted sentences without focus and added descriptions of key results.
Point 3: Keyword must in alphabtical order.
Response 3: We are very sorry for our negligence of this problem and we have adjusted the order of keywords properly.
Point 4: Introduction grammatical issues and lacking of clear thread.
Response 4: We are very sorry for the errors and we have re-examined the grammar and structure of the introduction and added relevant information to make the research purpose clearer by considering the Reviewer’s suggestion.
Point 5: The purpose and detail of the used variety need be supplemented.
Response 5: We are very sorry for our negligence of these problems. We have supplemented the relevant information of the collected samples.
Point 6: The figures are quite low resolution and difficult to make out.
Response 6: We are very sorry for our negligence of this problem. We have replaced higher quality pictures in the text and uploaded them again.
Point 7: The number of samples and repetitions are not clear.
Response 7: We are very sorry for our negligence of these problems. We have supplemented the specific methods of sample collection and analysis.
Point 8: The discussion should be interpreted with the results as well as discussed in relation to the present literature.
Response 8: Considering the Reviewer’s suggestion, we have improved the discussion and combined the relevant literature to discuss.
Point 9: Conclusion section is very lengthy. The author should emphasize this in a better way.
Response 9: Considering the Reviewer’s suggestion, the content of the conclusion has been streamlined and the focus has been highlighted.
Point 10: The whole References should be corrected.
Response 10: We are very sorry for the errors and the Reference has been modified according to the standard.

Round 2
Reviewer 1 Report
please see attached file.

Author Response
Response to Reviewer 1 Comments
Dear Reviewer:
I would like to express our sincere gratitude for your thoughtful and insightful reviews of our manuscript entitled “Analysis of the RNA Editing Sites and Orthologous Genes Function of Transcriptome and Chloroplast Genome and the Evolution of Five Deutzia Species” (ID: IJMS-2249135). Your feedback has been invaluable in shaping my work and improving the quality of my research. We have carefully studied all the comments and suggestions and made the necessary revisions to the best of our ability to improve the clarity and quality of our work which we hope meet with approval. And the revised version was marked by “Track Changes” function. The main corrections in the paper and responds to the comments are as following:
Point 1: There are still typos and grammatical errors.
Response 1: We are very sorry for the errors in our writing. We have carefully gone over the spelling and grammar and corrected improper words.
Point 2: Non C-to-U type RNA editing sites with low sequencing reads need to be removed. And non C-to-U type editing needs additional experimental evidence. And the relevant parts need to be modified.
Response 2: Considering the Reviewer’s suggestion, we have deleted the sites with low sequencing depth in the text. The relevant parts were also revised.
Point 3: The pi value of flanking sequences needs analysis. And the figure which may mislead readers should be removed.
Response 3: We have analyzed the pi value of intergenic sequences according to the reviewer’s comments, and we have picked the sequences with relatively high pi, which might be used as molecular markers. Figure 3 was also removed.
Point 4: The “exchange” of chloroplast and nuclear was not accurate. The usage of “genes” and “proteins” was not clear. And the genes orthologous to chloroplast and ribosomal need to explain.
Response 4: We are very sorry for our negligence of these problems. We have corrected the description of gene transfer in a more accurate way. In the previous version, we confused “gene” and “protein” because GO annotations are annotated after converting CDS into amino acid sequences. And we have modified the relevant statements.
Once again, thank you very much for your comments and suggestions!

Reviewer 2 Report
Dear Editor,
Thank you for providing the opportunity to review the revised manuscript. The manuscript is improved considerably after revision according to the reviewer's comment. Now this study is a suitable contribution to the IJMS. I recommend the manuscript for publication.
Thank you
With best regards
Author Response
Response to Reviewer 2 Comments
Point 1: The manuscript is improved considerably after revision according to the reviewer's comment. Now this study is a suitable contribution to the IJMS. I recommend the manuscript for publication.
Response 1: We are grateful to the Reviewer for reviewing the paper so carefully. Once again, thank you very much for your comments and suggestion!

Round 3
Reviewer 1 Report
Regarding the issue of non C-to-U RNA editing, the authors have now removed all genes except atpF from the new Table 4 (formerly Table 3), but they include a new Figure 6 which creates an additional error. On p.12 they say “... introns in atpF (Fig. 6), which may disturb the identification of editing sites.” The chloroplast atpF gene in plants does have a group II intron at that position, but it is about 700 bp long in plant chloroplasts (see for example, Fig. 7 in Ostersetzer et al. Plant Cell 2005 or Fig.3 in Bird et al. EMBO J. 1985). By comparing Fig.6 with known chloroplast atpF sequences, it is evident that nucleotides shown in light blue in Fig.6 correspond to the first 12 nt of the intron, so the authors have either misinterpreted sequencing data or there is an experimental artefact. Chloroplast introns (group I and group II) have very well-defined structures required for splicing and there is no possibility that Deutzia has a 12 nt intron. Their comment “may disturb the identification of editing sites” is not valid - the chloroplast atpF gene and its intron/splice sites are very well-documented and highly conserved in plants, but depend on high quality sequence data and careful interpretation. Corrections are needed to this section - Figure 6 should be removed and it seems very likely that Table 4 is also not justified.
On p.11 the authors still say “34−39 editing events were identified through RNA-Seq mapping” although the Abstract had been revised to “a total of 32-33" in the previous ms. version. It seems likely after the atpF issue is resolved, the new number will be lower (and all will be conventional C-to-U as expected), and the comment about “most being C-to-U" will be removed from the Abstract and the Discussion section.
There are still many grammatical errors and problems with English usage, especially in the new sentences that have been added. Below are just a few examples.
p.3 “Therefore, better understand of the development, evolution and internal molecular mechanism of the leaves of Deutzia spp. is necessary”
p.3 “how integrated into the nuclear genome is still known little”
p.5 “TrnL-UAG was a tRNA gene and rps3 encoded proteins of small ribosomal subunit, and these two genes belonged to self-replication category. PetN belonged to photosynthesis category and encoded Subunits of cytochrome b/f complex.”
p.5 “might be useful for molecular marker or further phylogenetic analysis”
p.7 “It indicated that most genes were subjected to purification selection effect. And ccsA, matK, rbcL, and ycf1 with KA/KS > 1 provided evidence of positive selection”
Author Response
Response to Reviewer 1 Comments
Dear reviewer:
We appreciate the time and effort you dedicated to reviewing our paper entitled “Analysis of the RNA Editing Sites and Orthologous Genes Function of Transcriptome and Chloroplast Genome and the Evolution of Five Deutzia Species” (ID: IJMS-2249135). Your comments and suggestions has not only improved the quality of this paper but will also undoubtedly shape our future research endeavors. We assure you that we have carefully considered each of your comments and suggestions and have made the necessary revisions to address them. And the revised version was marked by “Track Changes” function. The main corrections in the paper and responds to the comments are as following:
Responds to the reviewer’s comments:
Point 1: The explanation of introns and editing sites of atpF gene is not accurate.
Response 1: According to the Reviewer’s suggestion, we have reconfirmed the DNA, cDNA and CDS sequences. We found that there were inaccuracy between the cDNA and CDS. This may because of the misidentification of gene structure in sequencing process. The intron was located after the missing 12nt. And we have modified the wrong part.
Point 2: The number of RNA editing sites should be corrected.
Response 2: We are very sorry for the negligence in our writing. We have corrected the number of RNA editing sites.
Point 3: There are still many grammatical errors and problems with English usage.
Response 3: We are very sorry for the errors in our writing and we have rechecked the grammar and made correction.
